# Detection of Transcription Factors Related to Axillary Bud Development after Exposure to Cold Conditions in Hexaploid *Chrysanthemum morifolium* Using *Arabidopsis* Information

**DOI:** 10.3390/plants12173122

**Published:** 2023-08-30

**Authors:** Tsuyoshi Tanaka, Katsutomo Sasaki

**Affiliations:** 1Research Center for Advanced Analysis, National Agriculture and Food Research Organization (NARO), 2-1-2 Kannondai, Tsukuba 305-8518, Ibaraki, Japan; 2Institute of Vegetable and Floriculture Science, National Agriculture and Food Research Organization (NARO), 2-1 Fujimoto, Tsukuba 305-0852, Ibaraki, Japan

**Keywords:** axillary bud, *Chrysanthemum morifolium*, RNA-Seq, transcription factor

## Abstract

Chrysanthemum is one of the most commercially used ornamental flowering plants in the world. As chrysanthemum is self-incompatible, the propagation of identical varieties is carried out through cuttings rather than through seed. Axillary bud development can be controlled by changing the temperature; for instance, axillary bud development in some varieties is suppressed at high temperatures. In this study, we focused on the simultaneous axillary bud growth from multiple lines of chrysanthemum upon changing conditions from low to normal temperature. Transcriptome analysis was conducted on the *Chrysanthemum morifolium* cultivar ’Jinba’ to identify the important genes for axillary bud development seen when moved from low-temperature treatment to normal cultivation temperature. We performed RNA-Seq analysis on plants after cold conditions in two-day time-course experiments. Under these settings, we constructed a transcriptome of 415,923 *C. morifolium* and extracted 7357 differentially expressed genes. Our understanding of *Arabidopsis* axillary meristem development and growth showed that at least 101 genes in our dataset were homologous to transcription factors involved in the biological process. In addition, six genes exhibited statistically significant variations in expression throughout conditions. We hypothesized that these genes were involved in the formation of axillary buds in *C. morifolium* after cold conditions.

## 1. Introduction

Chrysanthemum (*Chrysanthemum morifolium* Ramat.) is one of the most commercially distributed flowering plants in the world, and it has long been the top cut flower plant in Japan in terms of both volume and value. Both potted and cut flowers are used commercially, but cut flowers are the main commercial products in Japan and worldwide. Every year, many new varieties with different flower colors, petal shapes, cultivability, disease resistance, and other characteristics are produced mainly through cross-breeding. Chrysanthemum is basically self-incompatible. The genome is not homogenous, different from diploid plants, e.g., *Arabidopsis*, rice, and tomato, which are used as model plants in research. As a result, maintaining the traits of the parental lines of chrysanthemum in seed is almost difficult. Cuttings are used in chrysanthemum propagation and in the maintenance of identical cultivars.

Establishing a method for controlling the production and growth of desired axillary buds is important in chrysanthemum. In the case of disbudded chrysanthemum, which should be grown as a single flowerhead for commercial use, axillary buds are unnecessary because they reduce the commercial value of the variety, and therefore, axillary buds must be removed in the process of cultivation. On the contrary, in the case of chrysanthemum, it is critical to secure cuttings, i.e., to stimulate axillary bud development, in order to reproduce the same variety. Thus, control of axillary bud formation and growth is important because axillary buds are needed or not needed depending on the use of the chrysanthemum. Numerous reports on genes or plant hormones involved in axillary bud formation and growth have been found in rice and *Arabidopsis* [1,2,3].

Transcription factors (TFs) play a key role in signal transduction via binding with DNA sequences and regulating the expression of other genes. Consequently, changes in TF expression can impact downstream genes and influence various biological phenomena. In *Arabidopsis*, a variety of TFs have been identified as being critical for axillary bud formation and regulation, and the signaling hierarchy of each TF has been identified [3]. It has also been discovered that miRNAs regulate TFs necessary for axillary bud development [2]. Among the TFs that have been shown to be important for axillary bud formation in *Arabidopsis*, it has been reported that axillary bud formation is suppressed by the lateral suppressor-like gene [4] in an antisense transgenic chrysanthemum [5]. Thus, homologous TFs function as very important regulators of axillary bud formation across plant species.

Growing temperature is one of the most important factors influencing the growth of chrysanthemum, as observed in many other plants. Nevertheless, the effect of temperature on floral development in chrysanthemum differs based on their variety and is a complex process since floral development is suppressed only within a suitable temperature range for both high and low temperatures [6]. It is also known that different varieties have different responses to high and low temperatures and different tolerances of reaction temperatures [7,8]. In addition, in the case of chrysanthemum, when axillary buds are not formed or are difficult to form due to exposure to high temperatures during cultivation, cultivation at low temperatures for about one month will promote axillary bud development again, which will result in the ability to obtain cuttings and thus facilitate the propagation of the chrysanthemum cultivars. This is known to increase axillary bud development again, which aids in chrysanthemum propagation. Additionally, it is known that while maintaining and propagating sterile seedlings of chrysanthemum by regularly repotting individuals in a plant box for a long period of time in order to maintain or propagate the plants, a phenomenon in which growth becomes poor for some reason may be noticed. Sluggish growth caused by continued sterile culture of chrysanthemum may be improved by a one-month low-temperature condition.

Transcriptome analysis has been performed in a variety of plants, including non-model species, to better understand biological processes at the genetic level [9]. Although long-read sequence techniques enhanced the accuracy of transcript structure annotation [10], expression profiling was performed mainly using the Illumina platform. There were reports on transcriptome analysis in chrysanthemum [11,12]. In 2020, long-read sequencing was also used in transcriptome analysis [13]. However, because of its hexaploid and heterogeneous genomes, chrysanthemum transcriptome analysis is relatively behind that of other plants. Even in cereal crops, transcriptome analysis is more complicated in maize and wheat than in rice, and analytic tools for the solution have been developed [14,15]. Even though reference genome sequences exist for wild-type diploid species of *C. seticuspe* [16,17], sequence diversity between the target species and the wild species should be considered for statistical reliability.

Low-temperature stress in wild-type diploid *C. seticuspe* and hexaploid cultivated chrysanthemum has been analyzed using transcriptome analysis [18,19]. On the other hand, plant growth after low-temperature cultivation has been the subject of little research. In this study, we found that simultaneous axillary bud development could be visually observed after transfer from the low-temperature condition to room temperature. Hence, we investigated axillary bud development after low-temperature conditions using transcriptome analysis. As a result, gene expression changed significantly on the first day after transfer from low temperature to room temperature, and six TF genes important for axillary bud development varied among them.

## 2. Results

### 2.1. Sample Preparation for the Transcriptome Analysis for C. morifolium ‘Jinba’

Cuttings of uniform size were made from the parental lines of chrysanthemum grown in the greenhouse to propagate chrysanthemum plants under the same growing conditions (Figure 1a). The cuttings were put in the rooting medium and given a month in the greenhouse to grow roots. The fully rooted cuttings were replanted in 9 cm flowerpots and grown in the greenhouse for two months. A chrysanthemum approximately 25 cm in size was maintained at 4 °C for 3 weeks and then moved to a 25 °C plant-growth chamber. For sampling nodes for transcriptome analysis, the fully expanded leaves, as seen from the apex, were numbered from the top to the bottom (Figure 1b), and three replications of each node were sampled for each of the three lines, with zero time as the day immediately before the transfer from 4 °C to 25 °C and one and two days after the transfer, respectively. Axillary buds were checked visually (Figure 1c). At this time, axillary buds were not observed at nodes 1 and 2, only at node 3 (Figure 1d). Axillary buds were not visible at node 1 one day after the transfer. Two days after the transfer, axillary buds were observed at all nodes. Stems with axillary buds exposed, except for the petiole, were collected for transcriptome analysis (Figure 1e; stems in the region where the petiole was cut; red boxed area).

### 2.2. Construction and Evaluation of the Reference Transcriptome for C. morifolium ‘Jinba’

#### 2.2.1. RNA-Seq Data and Assembly

We targeted two positions for axillary bud development (first and second buds) over two days (0, 1, and 2 days after cold condition). In the case of 0 and 1 day, since there were no buds, we collected tissues from the roots of the branch. In total, we collected six different conditions from three independent individuals. RNA-Seq data from Illumina platforms contained 21,337,585–29,951,108 read pairs (Appendix A). After quality control by Trimmomatic, 97.65–98.49% of the raw reads remained (20,456,269–29,385,376). Trinity generated 1,007,492 isoforms corresponding to 415,923 genes, which accounted for 92.06% NGS reads (Table 1). The total length of these isoforms was 634,179,802 bp. However, when we attempted reference-based RNA-Seq analysis using reference genomes *of C. seticuspe*, less than 80% of the NGS reads could be mapped. As a higher mapping ratio of NGS reads was achieved using Trinity genes than using *C. seticuspe* genomes, we decided to use Trinity data for subsequent analysis.

#### 2.2.2. Evaluation of Redundancy and Annotation of ‘Jinba’ Transcriptome Data

Although our transcriptomic data exhibited high NGS read coverage, we observed that 57.36% of these reads were concordantly aligned two or more times. Despite ‘Jinba’ being hexaploid, the number of genes and isoforms was higher in ‘Jinba’ than in *C. seticuspe*, potentially leading to redundancy. In fact, the relationship between genes and isoforms revealed that 137,814 genes (33.1%) encompassed multiple isoforms. Among these, 19,749 genes contained 10 or more isoforms (maximum isoforms: 49).

We clustered these genes using two different methods. First, we conducted de novo sequence clustering using EvidentialGene and constructed 325,337 representative transcripts. However, mapping NGS reads to these representative sequences showed only 5% higher unique hits than the Trinity transcriptome, despite the low total mapping ratio (70.72–72.94% for each sample). Therefore, we concluded that the representative sequences were unsuitable for this study.

Second, we performed comparative genetics analysis of *Arabidopsis* and *C. seticuspe* proteome. Homology-based sequence clustering using *Arabidopsis* and *C. seticuspe* proteomes showed that our transcript isoforms covered 69.6% of the *Arabidopsis* genes and 71.5% of *C. seticuspe* genes, of which 51.6% and 47.5% were with more than 50% alignment coverage (Table 1). From the viewpoint of our dataset, while 34.0% had sequence homology to both *Arabidopsis* and *C. seticuspe* genes, 44.7% had no hits in either gene set. At the gene level, 229,016 of 415,923 were not assigned to any genes from *Arabidopsis* or *C. seticuspe* (Appendix A).

We also evaluated the transcriptome data based on the coding capacity. We used a transdecoder to predict 350,636 longest ORFs from all isoforms, which covered 111,592 of 415,923 genes. Of the 304,331 genes without predicted ORFs, 202,894 did not have any sequence homology against *Arabidopsis* genes or *C. seticuspe* genes, and these genes were either fragmented or low-quality genes. The distribution of isoforms showed that 50.0% of isoforms without ORFs or sequence homology were less than 200 bp in length, whereas only 79.0% of isoforms with any biological annotation were more than 200 bp in length.

We also evaluated the genes based on the expression level using counts per million (CPM), calculated from the Trinity and Salmon programs. In this study, we counted reads for each isoform and calculated the CPM for each gene using R programs. First, only 72,533 of 415,923 genes had >1 CPM for three replicates, at least in one condition. This meant that 73.6% of all genes had a low expression, observed by accident, in only one or two replicates. By integrating the data with a homology search, we found that 60,969 of 72,533 genes had any sequence homology to *Arabidopsis* and *C. seticuspe* genes. As a result, we were able to confirm that expressed genes had biological functions that were annotated by sequence homology.

### 2.3. Detection of Differentially Expressed Genes under Axillary Bud Formation in C. morifolium ‘Jinba’

#### 2.3.1. Differentially Expressed Genes among the Six Conditions

We detected differentially expressed genes (DEGs) by the statistical test of edgeR with Trinity/Salmon outputs. First, we targeted the above mentioned 72,533 genes with CPM > 1 for three replicates in at least one condition, while all genes were used for normalization. We found that samples from 0 days were not clustered between nodes 1 and 2, whereas samples from day 1 and day 2 were clustered between nodes 1 and 2, respectively (Figure 2). This clustering suggested that gene expression at 0 days between nodes 1 and 2 was more different than that at 1 or 2 days, and the gene expression profile would be synchronized promptly between node 1 and 2 after recovering from cold conditions. In addition, since the gene expression was different between 1 day and either 0 days or 2 days, DEGs might be targeted in 1 day.

Next, we performed an exact test for pair-wise comparison of six conditions by edgeR and found 37,263 DEGs with a threshold of FDR < 0.05 (25,717 with FDR < 0.01) in any combination of two conditions. Spearman’s correlation heatmap revealed that 0-day (control) samples were clustered separately in axillary nodes 1 and 2, while 1-day and 2-day samples were clustered for the group (Appendix A). The difference in the topology observed between Figure 2 and Appendix A was speculated by clustering methods.

#### 2.3.2. Annotation of DEGs Based on Arabidopsis Annotation

We annotated DEGs using *Arabidopsis* data because of the huge amount of annotation data with experimental evidence. Using the Database for Annotation, Visualization, and Integrated Discovery (DAVID) (https://david.ncifcrf.gov/gene2gene.jsp, accessed on 1 September 2022) [20], we performed functional annotation of 37,263 DEGs from *C. morifolium* ‘Jinba’ in this study using 16,877 AGIs assigned to the DEGs. To begin, nearly all AGIs were assigned to a gene ontology (GO) term. A total of 15,474 AGIs were classified as “Cellular Component”, whereas 11,719 and 9947 were classified as “Biological Process” and “Molecular Function”, respectively. Oppositely, only 3044 AGIs could be assigned to any Kyoto Encyclopedia of Genes and Genomes (KEGG) pathway. This result is linked to only three enriched “Biological Process” terms for metabolism (Benjamini adjusted *p*-value < 0.05): “negative regulation of metabolic process, fucose metabolic process, and negative regulation of cellular metabolic process”. In “Biological Process” we also found that the GO terms “protein phosphorylation” for signal transduction, “cell division” for axillary bud development, and “response to cold” for recovering from cold conditions were enriched. Therefore, we considered that the AGIs were reliable in this study.

DEGs expressed under certain conditions were quite important for the axillary bud’s development. By comparing samples among the same tissues (either node 1 or node 2), we curated DEGs in a time-course manner. Then, 19,024 and 20,089 DEGs showed significant expression changes from day 0 to 2 (Figure 3). Day-specific DEGs were the highest in the day 1 condition for both node 1 and 2. Moreover, the time course from day 0 to 1 showed the most expression changes in both nodes 1 and 2. When the DEGs in nodes 1 and 2 were integrated, only 7357 genes were candidates. While 0- and 2-day-specific DEGs were drastically reduced (2.4–41.1%), 1-day-specific DEGs remained (27.9–77.2%). This similarity of DEGs between nodes 1 and 2 distributions is shown in the dendrogram (Figure 2). Time-course DEGs comparisons showing that expression levels in 1 day were significantly changed suggest that axillary bud development after cold conditions started on day 0 to day 1. Based on the DEGs in 1 day (2693 for upregulated DEGs and 481 for downregulated DEGs), functional analysis was carried out using DAVID. A total of 1655 and 278 AGIs from upregulated and downregulated DEGs, respectively, were used for the analysis (Appendix A). GO analysis for “Biological Process” using upregulated DEGs showed that “protein phosphorylation” and “response to cold” were listed but “cell division” was left off the list. In addition, terms relating to the immune system and defense were observed. GO terms from downregulated DEGs were related to chromosome compositions, e.g., “nucleosome assembly”, “chromatin silencing”, and “chromosome condensation.” Even if we need to evaluate the reliability of experiments, DEG analysis separated by expression profiles of upregulation/downregulation for sample comparisons would be more focused on the direct biological process.

#### 2.3.3. Expression Profiles of TFs Relating to Axillary Bud Formation

In *Arabidopsis*, there are reports of TFs relating to axillary meristem formation and growth [21,22]. Using this knowledge, we analyzed the expression profiles of TFs in *C. morifolium* ‘Jinba’. We listed 31 TFs and assigned 101 ‘Jinba’ genes as homologous to 26 *Arabidopsis* TFs (Appendix A). A dendrogram of 101 TFs showed that 10 genes were separately clustered: “TRINITY_DN100544_c3_g1” (identity = 48.7%); “TRINITY_DN123887_c1_g1” (44.3%); “TRINITY_DN122473_c0_g2” (62.6%); “TRINITY_DN97218_c2_g1” (38.4%); “TRINITY_DN127587_c0_g2” (66.9%) homologous to At-FUL/AGL8 (MADS); “TRINITY_DN129352_c3_g2” (46.8%), “TRINITY_DN87924_c0_g1” (39.8%) homologous to AtRAX1 (MYB); “TRINITY_DN117535_c0_g1” (44.2%); “TRINITY_DN117535_c0_g2” (25.5%) homologous to AtAS1: ATMYB91 (MYB); and “TRINITY_DN100783_c1_g1” (62.7%) homologous to AtSTM (KNOX) (Figure 4). Their expression patterns were upregulated on day 1, especially those of bud 1, and downregulated on day 2. Since other homologues of At-FUL/AGL8 (MADS), AtRAX1 (MYB), and AtAS1: ATMYB91 (MYB), except for AtSTM (KNOX), existed in *C. morifolium* ‘Jinba’, we analyzed the relationship between the expression profiles and sequence conservation in *Arabidopsis* homologues. By pair-wise comparison, we calculated sequence identity (based on *Arabidopsis* genes). In the case of AtAS1: ATMYB91 (MYB) homologues, the focused transcripts had higher identities than the other homologue, “TRINITY_DN117535_c0_g2” (17.2%). On the other hand, two transcripts homologous to AtRAX1 (MYB) were ranked second and third among the six homologues. The homologue with the highest identity was “Trinity_DN98932_c1_g3” (72.8%), and the expression profile showed the more moderate changes on 1 day for node 1. If we assumed that the conserved expression profiles between *Arabidopsis* and chrysanthemum were followed by sequence conservation, the expression profiles of AtAS1: ATMYB91 and AtRAX1 (MYB) during axillary bud development were different. In the case of At-FUL/AGL8 homologues, the rank of sequence conservation was nested (first, third, sixth, seventh, and ninth) for homologues in the separated cluster. In our dataset, 11 At-FUL/AGL8 homologues were composed of 107 isoforms, and this complex composition of the genes might be caused by the diversified expression profiles.

Combined with the DEG analysis, we had 36 homologues to 17 *Arabidopsis* TFs. Six homologues showed 1-day-specific DEG, and the profiles were completely opposite between two transcripts (upregulated), “TRINITY_DN101946_c1_g1” and “TRINITY_DN100109_c0_g1” homologous to AtWRKY71/EXB1 (WRKY) and AtEBE (AP2/ERF), respectively, and four transcripts (downregulated), “TRINITY_DN117535_c0_g1” and “TRINITY_DN117535_c0_g2” homologous to AtAS1: ATMYB91 (MYB) and “TRINITY_DN107968_c0_g2” and “TRINITY_DN89998_c0_g1” homologous to AtLOB1 (LOB). These data suggested that TF expression variations on day 1 were signals for axillary bud formation following cold conditions. To validate the results of in silico analysis, we performed experimental expression analysis. We designed primers for six chrysanthemum genes that were orthologous to *Arabidopsis*, namely TFs, EBE, EXB1, AS1-1, AS1-2, and LOB (Appendix A). As there are two DEGs for LOB orthologs, we constructed two kinds of primers for each gene (here, we describe the two LOB genes as LOB-1 and LOB-2). Subsequently, we performed quantitative real-time reverse transcription polymerase chain reaction (qRT-PCR) (Figure 5). Although the expression profiles obtained using qRT-PCR did not always match with those predicted in the in silico analysis, we observed changes in the expression levels of all five genes within one day. This finding suggests that the signal transduction pathway involved in the recovery of bud formation in *Arabidopsis* after cold conditions was initiated between days 0 and 1.

## 3. Discussion

As discussed in the Introduction section, there are many reports on the transcriptome analysis of *C. seticuspe* [18] and chrysanthemum [11,12,19], and many genes of chrysanthemum may be related to its growth and development. For example, the basic region/leucine zipper (bZIP) genes negatively regulate shot branching in chrysanthemum [23]. Reports focused on axillary bud formation have extensively studied TFs related to auxin and cytokinin pathways [24,25,26]. These transcripts, however, were created under specific conditions, such as florets, capitula, and flavonoid biosynthesis. Even in the study of axillary bud formation, the difference in the experimental conditions altered the relating TFs. In addition, no reports have published assembled transcript datasets. As this study aimed to determine the key genes for axillary bud development after exposure to cold conditions, expressed genes in the tissue under these conditions were targeted.

Our transcriptome data were obtained from high-quality Illumina reads (Table 1). However, to optimize the use of NGS data, we explored several RNA-Seq approaches. First, we evaluated the feasibility of using closely related reference genomes from *C. seticuspe* [16,17]. However, owing to the low mapping rate, we did not use this method. Subsequently, we conducted de novo transcriptome assembly that generated millions of isoforms from 425,923 genes. While 325,337 consensus sequences were generated using EvidentialGene [27], mapping results of RNA-Seq data showed a decreased mapping ratio. Since we considered that the low mapping ratio was worse than the increased multiple hit rate, we did not use the consensus sequences in this study. We also hypothesized that the diversity of the allele sequence due to hexaploidy in *Chrysanthemum* could have contributed to the low mapping rate. Therefore, it is also possible that functional diversification occurred between alleles. This was observed in DEG analysis, which revealed differential expression profiles in transcripts of certain genes, such as AtAS1: ATMYB91 and AtRAX1 (MYB). Future studies should focus on performing a more accurate expression analysis for alleles.

We also evaluated the transcriptomic data based on the coding capacity. However, at that time, we could not annotate 232,390 of 415,923 chrysanthemum genes without any sequence homology to any *Arabidopsis* or *C. seticuspe* genes. From the results of ORF prediction and sequence length calculations, we confirmed that the reliability of the genes was low. However, we also understood that there were chrysanthemum genes in the biological process. In addition to DEG analysis, biological annotation and functional validation will be performed in further studies.

DEG analysis raised 7357 possible candidates relating to axillary bud generation by combining the results from nodes 1 and 2. We considered that this extraction would extract commonly relating genes for axillary bud development and discard noised DEGs from the contaminated tissues by manual sampling. Of 7357 DEGs, more than half of the genes were observed in 1 day (Figure 3). This means that expression changes in the axillary bud development occurred from day 0 to day 2. The results are consistent with the phenotypic observation in Figure 1, in which axillary buds were observed from day 1. If we focus on the primary event of axillary bud development, a shorter time course should be planned (between 0 and 1 day).

Since we considered that signal transduction is the primary event for axillary bud development and affects largely downstream genes, we targeted the TFs. We annotated 101 *Chrysanthemum* genes as TFs associated with axillary meristem formation and growth based on their sequence similarity with 31 *Arabidopsis* genes [3]. Although in silico analysis revealed six DEGs, the experimental validation through qRT-PCR yielded slightly different results. Although further studies should perform a more accurate validation of gene expression, we observed the timing of expression changes while *Chrysanthemum* recovered from cold conditions, which occurred between days 0 and 1. This finding aligns with the phenotypic observations (Figure 1), and a shorter time-course experiment would be beneficial for comprehending axillary bud formation via signal transduction.

Owing to the presence of numerous uncertain DEGs in the current in silico analysis, we focused solely on TFs. However, we used the transcript data to target TFs involved in the biological response and designed probes for experiments. Both in silico analysis and experimental results showed that expression changes in TFs were initiated on day 1, prompting us to focus on the earlier stages of bud formation in our study.

The DEGs selected in this study included EXB1 (TRINITY_DN101946_c1_g1), EBE (TRINITY_DN100109_c0_g1), AS1 (TRINITY_DN117535_c0_g1 and TRINITY_DN117535_c0_g2_i1), and LOB (TRINITY_DN89998_c0_g1 and TRINITY_DN107968_c0_g2). Their orthologs selected as DEGs in this study were located in the signaling pathways for axillary meristem formation (Figure 6a) and growth (Figure 6b) [21,22]. EXB1 and EBE were located in the most upstream regions of their respective signaling pathways, and our experiments indicated that gene expression levels were upregulated in 1 day. Therefore, we postulated that axillary bud formation commences between days 0 and 1 after recovering from cold conditions. Notably, our qRT-PCR findings differed from those obtained while analyzing *Arabidopsis*, such as in terms of LOB. We also observed variation in detailed expression profiles obtained using in silico RNA-Seq analysis and qRT-PCR, which may be attributed to the functional diversity of alleles due to hexaploidy in *Chrysanthemum*.

In this study, we could have either used reference-based or de novo RNA-Seq analysis to analyze gene expression and screen TFs. Reference-based RNA-Seq generally exhibited good outcomes, owing to reduced erroneous transcript assembly. However, we performed de novo transcriptome analysis without sequence clustering due to the following observations. First, <80% of NGS reads could not be mapped to the genome of the *Chrysanthemum* species [16,17]. We recognized that the absence of one-fifth of the data could affect the statistical tests used for screening DEGs. Second, we attempted sequence clustering to construct a representative transcriptome for ‘Jinba’ using an ordinal program. However, this also showed suboptimal NGS read coverage. We considered that a high mapping rate (data usage) should be prioritized in this study; therefore, we used all transcript data obtained from Trinity.

The choice of annotation data for this study also required meticulous consideration. We generated lists of homologous genes in the transcriptomes of ‘Jinba’, *Arabidopsis*, and *C. seticuspe*. Among the 415,923 genes in the ‘Jinba’ transcriptome, 229,016 lacked homology, with two annotation data from two species. ORF prediction and transcriptome length analyses indicated these genes as erroneous products. However, since the partial removal of transcriptome data generated erroneous statistical test results, we retained these transcripts for further analysis. Therefore, our dataset contained 415,923 genes in ‘Jinba’ transcriptome, which is six times the size of the *C. seticuspe* proteome. To validate these genes and construct comprehensive representative genes, additional analysis combining other transcriptome data is warranted.

To screen potential TFs associated with axillary bud development after cold condition exposure, we could pursue annotation data from either *Arabidopsis* or *C. seticuspe*. Regarding this, we considered *Arabidopsis* to be a better reference for the annotation of *Chrysanthemum* transcripts than *C. seticuspe*. The enrichment of biological data of *Arabidopsis* compensated for the closer evolutionary relationship of *C. seticuspe* with *Chrysanthemum*. Our data also indicated that the similarity in the sequence of *Chrysanthemum* with *Arabidopsis* and *C. seticuspe* was comparable in coding regions, as *Chrysanthemum* transcripts covered approximately 70% of the two annotation datasets. Furthermore, although we attempted functional annotation of *C. seticuspe* data through various datasets, such as InterPro domains and GOs, the identification of specific TFs based on these annotation data posed challenges owing to the lack of experimental evidence. For example, regarding LOB genes examined in this study, the proteome of *C. seticuspe* comprised 45 LOB domain–containing genes, whereas the transcriptome of ‘Jinba’ had 70 genes with 203 isoforms as homologs. As experimental evidence was inevitable for further screening of candidates, we initially used *Arabidopsis* annotation data. Through manual curation of published manuscripts, we compiled a list of 101 TFs associated with axillary meristem formation and growth. This list was subsequently combined with the DEG analysis, leading to the identification of six prospective TF candidates. The differential expression of the genes was validated through qRT-PCR.

We plan to analyze ‘Jinba’ transcriptome in further analysis based on the results obtained in this study. First, we will evaluate the reliability of this transcriptome data. By combining RNA-Seq data from various conditions and tissues, we intend to construct a comprehensive transcriptome dataset for chrysanthemum. This dataset will pave the way for performing comparative genomics analysis with *C. seticuspe*, offering insights into the effect of polyploidization. Second, we will conduct functional assessment of the targeted TFs using transgenic lines. This experiment will clarify the functional diversification of these TFs arising from speciation and polyploidization, subsequent to the divergence between *Chrysanthemum* and *Arabidopsis*.

## 4. Materials and Methods

### 4.1. Plant Materials

*C. morifolium* ‘Jinba’ cultivar was used in this study. Chrysanthemum plantlets were cultivated in a greenhouse. A plant-growth chamber was used under long-day conditions (16 h/8 h, day/night) for the low-temperature condition (4 °C) and subsequent 25 °C condition.

### 4.2. RNA-Preparation and Sequencing

Nodes of chrysanthemum (Figure 1e) for RNA purification were immediately frozen in liquid nitrogen and stored at −70 °C. Total RNA was extracted from the nodes using TRIzol (Thermo Fisher Scientific; https://www.thermofisher.com/jp/en/home.html), and the isolated toral RNAs were further purified using the Rneasy mini kit (QIAGEN; https://www.qiagen.com/jp/). The purified RNAs were used for RNA sequencing analysis (NovaSeq6000, 2 × 150 bp paired-end) (Illumina; https://jp.illumina.com/ accessed on 22 August 2021). NGS data were deposited in DDBJ DRA (BioSample: SAMD00589521-SMD00589538).

### 4.3. Construction of De Novo Transcriptome Assembly and Expression Analysis

Illumina RNA-Seq data were preprocessed by trimmomatic-0.36 to discard low-quality and adapter sequences (-phred33 ILLUMINACLIP: adapter.fa:2:30:10 LEADING:15 TRAILING:15 SLIDINGWINDOW:4:15 MINLEN:32) [28]. Then, de novo transcriptome assembly was performed using Trinity-v2.5.1 (--seqtype fq –SS_lib_type RF) [29]. Representative sequences of ‘Jinba’ transcriptome were constructed by EvidentialGene (VERSION 20 January 2022) [27]. Mapping of Illumina reads to the representative sequences was performed using hisat2–2.0.5 (--min-intronlen 20 –max-intronlen 10,000 –dta –rna-strandness RF) [30]. The mapping ratios to the consensus sequences were calculated for each RNA sample.

To estimate the expression level, we used two Perl programs in the Trinity package (align_and_estimate_abundance.pl –seqType fq –SS_lib_type RF –est_method salmon –trinity_mode –prep_reference, and abundance_estimates_to_matrix.pl –est_method salmon). Lastly, we used a Perl program from the Trinity package to find DEGs (DifferentialExpression/run_DE_analysis.pl —matrix salmon.gene.counts.matrix —method edgeR).

### 4.4. Annotation and Comparative Analysis of C. morifolium ‘Jinba’ Transcriptome

We downloaded annotation data for *Arabidopsis* (TAIR10) and *C. seticuspe* (‘Gojo’) from TAIR (https://www.Arabidopsis.org/ accessed on 21 April 2014) and PlantGarden (https://plantgarden.jp/ja/index accessed on 13 August 2021). A sequence homology search was performed by analyze_blastPlus_topHit_coverage.pl in the Trinity package.

Functional annotation for GOs and KEGG pathways was conducted using DAVID (https://david.ncifcrf.gov/gene2gene.jsp accessed on 1 September 2022) [20]. From sequence similarity, AGI codes were assigned to *the C. morifolium* ‘Jinba’ transcriptome. A total of 31 *Arabidopsis* TFs relating to axillary meristem formation and growth were extracted from previous reports [21,22]. By combining the homology search results, 101 *C. morifolium* ‘Jinba’ transcripts were identified as TFs involved in axillary bud development. Pair-wise alignments were constructed by mafft-7.475 with default parameters [31], and identity was calculated by a Perl script.

### 4.5. qRT-PCR Analysis for Chrysanthemum TFs

We conducted qRT-PCR analysis using a previously described method [29]. Total RNA was extracted from each sample (Figure 1d) using the Rneasy Mini Kit (Qiagen). cDNAs were synthesized from the extracted total RNA using the ReverTra Ace qPCR RT Kit (TOYOBO). qRT-PCR was performed using GeneAce SYBR^®^ qPCR Mix α No ROX (NIPPON GENE; https://www.nippongene.com/english/index.html), and signals were detected using the Thermal Cycler Dice^®^ Real Time System TP800 (TaKaRa) following the manufacturer’s instructions. The specific primer sequences for each gene are presented in Appendix A.

## 5. Conclusions

We identified 415,923 genes that were expressed in the axillary buds after cold conditions. A total of 7357 DEGs were candidate genes relating to axillary bud development. In our transcriptome, there are 101 homologues to *Arabidopsis* TFs relating to budding. Expression changes during the recovery from the cold condition were initiated between days 0 and 1 for the six tested genes. This study highlights the following key findings: (1) de novo transcriptome analysis was suitable for studying *Chrysanthemum* in the present context; (2) transcriptome data could support the design of probes to target specific genes in *Chrysanthemum*; and (3) the signal transduction pathway involved in axillary bud formation after recovery from the cold condition was initiated between days 0 and 1.

## Figures and Tables

**Figure 1 plants-12-03122-f001:**
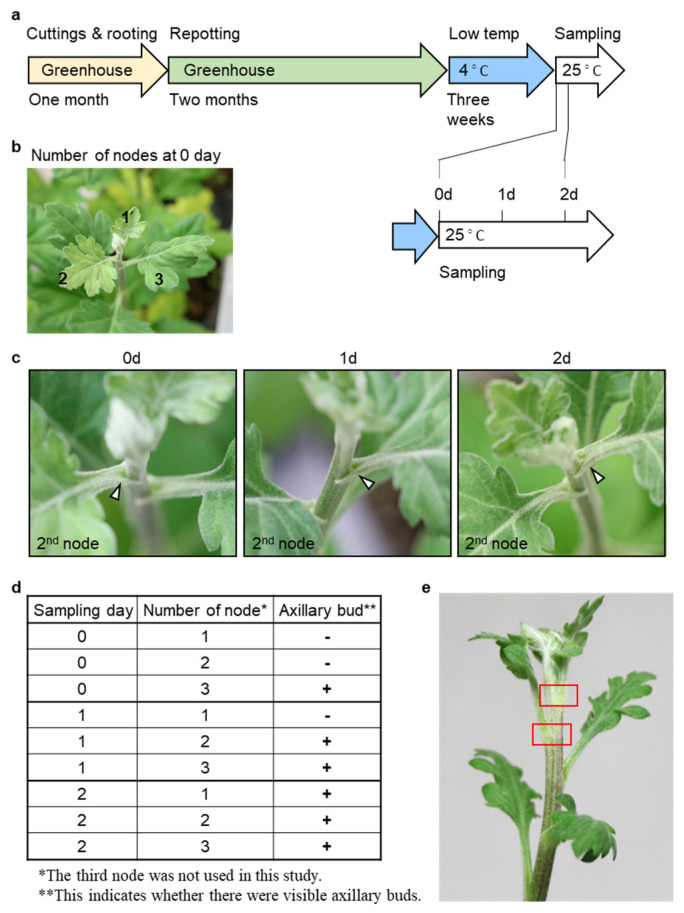
The preparation of samples for transcriptome analysis in chrysanthemum. (**a**) Illustration of plant growth and sampling process for transcriptome analysis. (**b**) Numbering of sampled nodes. Each node number was added to the leaves. (**c**) Confirmation of axillary bud development after the transfer from low-temperature condition to 25 °C. In the second node, the formation of axillary buds was detected at 0, 1, and 2 days after the transfer to 25 °C. (**d**) Axillary buds were visually observed at nodes 1, 2, and 3 at 0, 1, and 2 days after the transfer from 4 °C to 25 °C. (**e**) Sampled node positions (in red box). After the removal of the petioles, stems bearing axillary-bud-development sites were collected (the chrysanthemum plants in (**e**) are an example image and not used in this analysis).

**Figure 2 plants-12-03122-f002:**
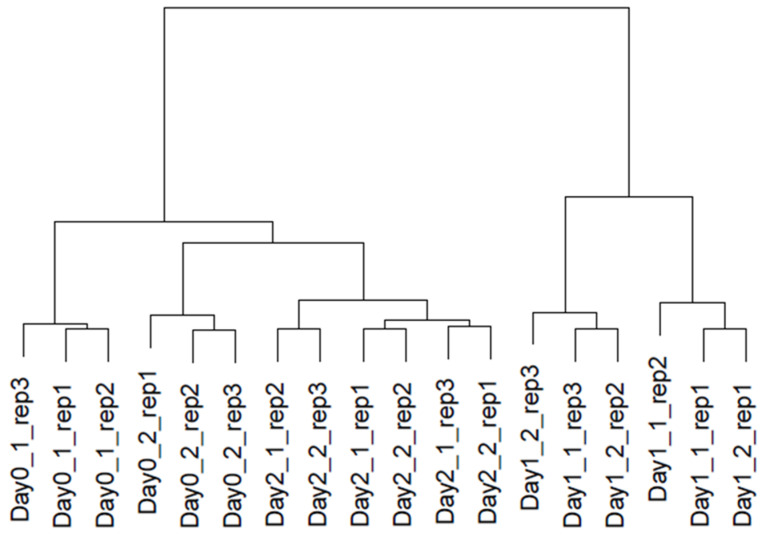
Dendrogram of 18 samples by expression patterns of 72,533 genes.

**Figure 3 plants-12-03122-f003:**
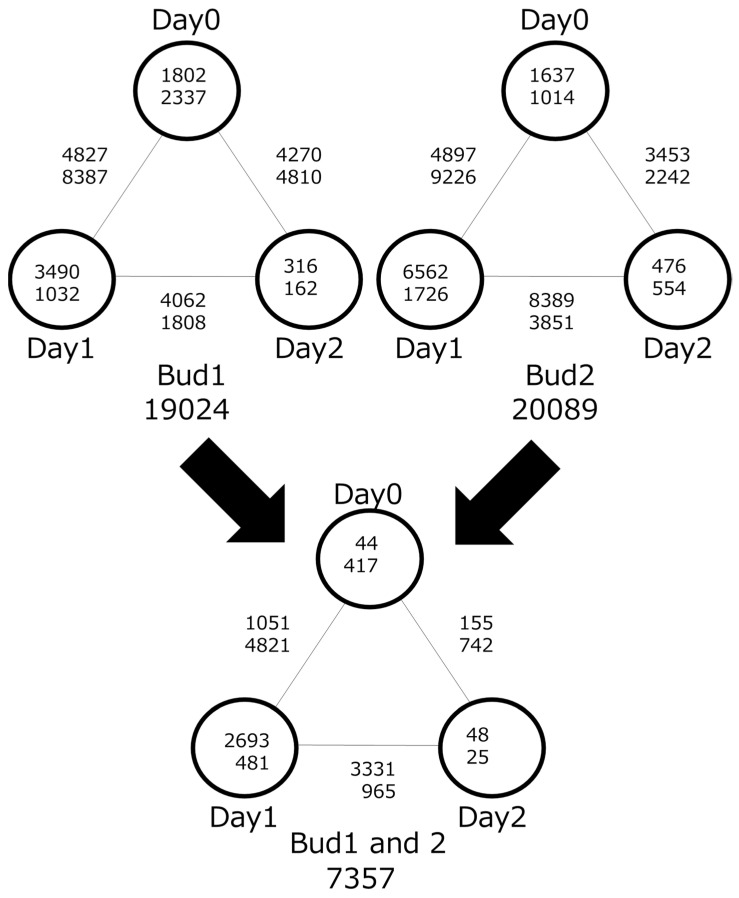
Identification of differentially expressed genes (DEGs) using time-course comparison in nodes 1 and 2. The top and bottom values show upregulated and downregulated genes during the early and late day, respectively. Values in circles represent condition-specific DEGs (specifically expressed in 0, 1, or 2 days). Values on lines represent DEGs from the comparisons between two conditions (0 day vs. 1 day, 0 day vs. 2 day, and 1 day vs. 2 days).

**Figure 4 plants-12-03122-f004:**
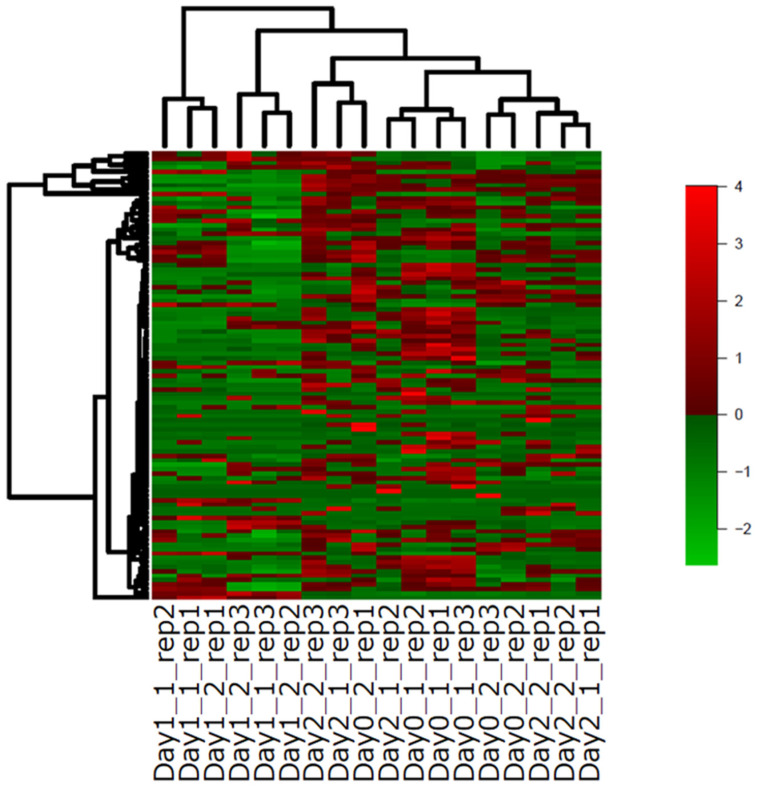
The heatmap of 101 DEGs with FDR < 0.05 among the six conditions based on the gene trimmed mean of M values.

**Figure 5 plants-12-03122-f005:**
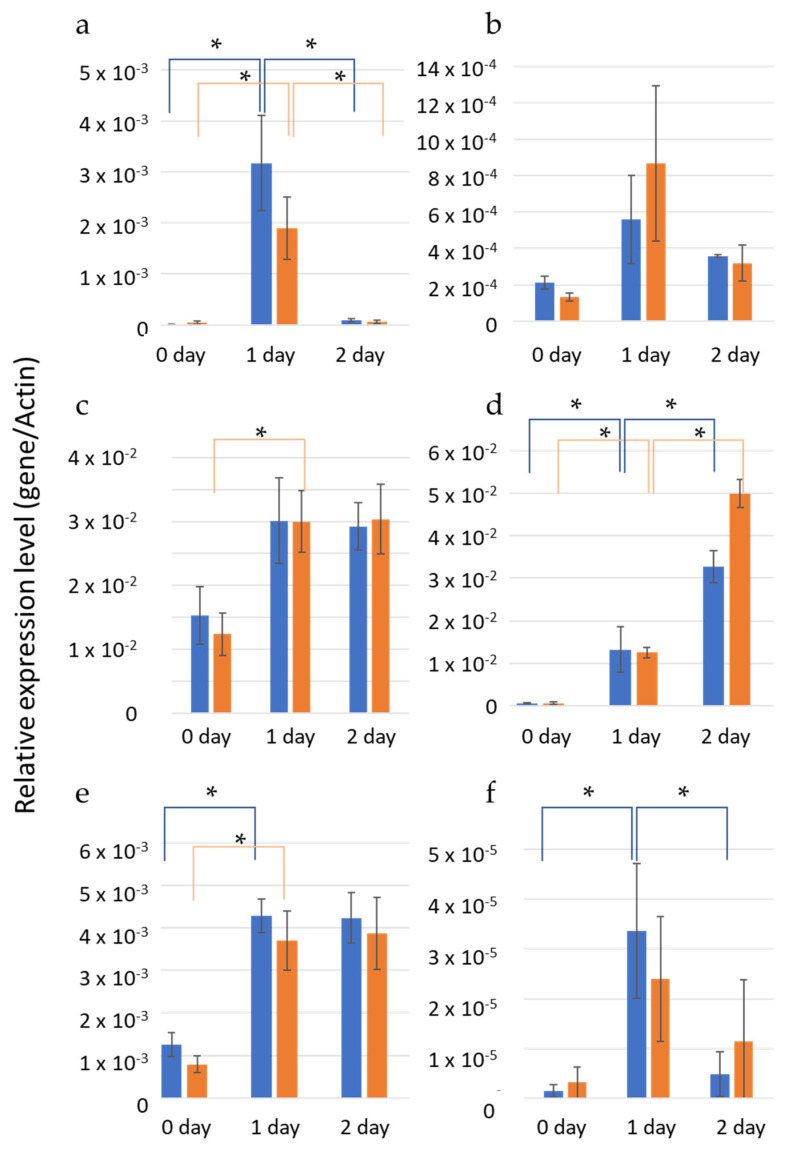
Expression profiles of six chrysanthemum genes: (**a**) EXE, (**b**) EXB1, (**c**) LOB-1, (**d**) LOB-2, (**e**) AS1-1, and (**f**) AS1-2 observed in bud 1 (blue) and bud 2 (orange) over 2 days. Y-axis represents relative expression level against Actin gene. Asterisks indicate statistically significant differences based on the *t* test (*p* < 0.05) when comparing the results obtained on days 0 and 1 or days 1 and 2.

**Figure 6 plants-12-03122-f006:**
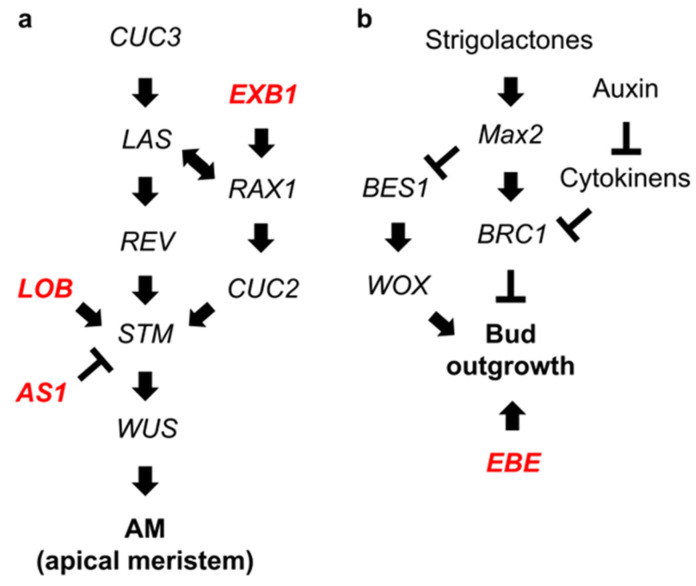
The diagram of signal transduction in regulating axillary meristem formation and growth, prepared in reference to that in *Arabidopsis*. Schematic of signal flow in regulating axillary meristem formation (**a**) and growth (**b**). Red letters indicate genes shown in the expression profiles in Figure 5 of this study.

**Table 1 plants-12-03122-t001:** Statistics of transcriptome assembly.

Data	Numbers
Isoforms	1,007,492
Genes	415,923
Average length of the isoforms (bp)	629
Genes with homology to *Arabidopsis* or *C. seticuspe* genes	186,907
Coverage of the transcriptome	
of *Arabidopsis*	19,074 (69.6%)
of *C. seticuspe*	53,071 (71.5%)
Clusters of isoforms by EvidentialGene	325,337

## Data Availability

The datasets generated during this study are mainly contained in Appendix A, and NGS data obtained in this study were deposited in DDBJ DRA (Bio Sample: SAMD00589521-SMD00589538).

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
