# Peer review of "Detection of Transcription Factors Related to Axillary Bud Development after Exposure to Cold Conditions in Hexaploid Chrysanthemum morifolium Using Arabidopsis Information"

_plants, 2023, doi:10.3390/plants12173122_

Round 1
Reviewer 1 Report (New Reviewer)
1. The paper is good. So my suggestion that it can be accepted.
2. In the future, I hope they should be test again and use in the greenhouse for applications.
Author Response
Dear. reviewer 1 Aug, 18th, 2023
Thank you very much for your very favorable decision.
Your decision is an encouragement for our research.
Thank you very much.
Sincerely yours,
Katsutomo Sasaki, Ph. D.
Senior Researcher
-----------------------------------------------------------------------------------------
NARO Institute of Floricultural Science (NIFS), National Agriculture and Food Research Organization (NARO), Fujimoto 2-1, Tsukuba, Ibaraki 305-8519, Japan
*E-mail: [email protected], Tel: +81-29-838-6822, Fax: +81-29-838-6841
-----------------------------------------------------------------------------------------

Reviewer 2 Report (New Reviewer)
The manuscript used the transcriptome and RT-PCR methods to identify 6 transcription factors associated with Chrysanthemum morifolium axillary bud development after cold treatment. This is too simple and crude.
- However, key experimental evidence is lacking. For example, CmLOB was heterologous expressed in Arabidopsis thaliana to observe the phenotype of its axillary buds.
In addition, the presentation of experimental results in this paper is difficult to understand and needs great improvement.
Therefore,it is not yet ready for publication, I must decline the present version of the manuscript.

No comments.
Author Response
Dear. reviewer 2 Aug, 18th, 2023
We will answer in red below after your suggestion or identification.
The manuscript used the transcriptome and RT-PCR methods to identify 6 transcription factors associated with Chrysanthemum morifolium axillary bud development after cold treatment. This is too simple and crude.
- However, key experimental evidence is lacking. For example, CmLOB was heterologous expressed in Arabidopsis thaliana to observe the phenotype of its axillary buds.
Response: We fully agree with the reviewer #2 comment. We found at least 70 genes with 203 isoforms were homologous to LOB-domain containing genes in Gojo proteomes. Therefore, to understand the biological function of each genes, we should conduct transgenic experiments either in Arabidopsis and Chrysanthemum. On the other hand, we screened six TFs as possible regulators for the phenotypes from in RNA-Seq analysis and validated the differentially expression of six genes in qRT-PCR. We will conduct RNAi and overexpression of the genes to know the exact biological efforts in the future work including the specific CmLOB which was identified by our screening in this study among the approximately 70 CmLOB genes (203 isoforms).
In addition, the presentation of experimental results in this paper is difficult to understand and needs great improvement.
Response: Thank you for your critical point for the manuscript construction. We modified explanation of results and discussion section for the better understanding of this manuscript for readers. We also added explanation sentences to figure/table legends based on your suggestions.
Therefore, it is not yet ready for publication, I must decline the present version of the manuscript.
On the PDF file of the manuscript
Whether it is abbreviated as TF, please keep it consistent throughout the manuscript.
Response: we changed all “transcription factors” to “TFs”.
The illustration should clearly state the meaning of the detail information in each part. For example, the numbers in the circles and the numbers between the circles.
Response: We appreciate your important suggestion. We added the detailed explanation in the legend of Figure 3 as below:
“Values in circles show condition specific DEGs (specifically expressed either in 0 day, 1 day or 2day) and values on lines show DEGs from the comparisons between two conditions (0 day vs 1 day, 0 day vs 2 day and 1 day vs 2 day).”
The colour legend should be added.
Response: We added the scale bar for the explanation.
What does the Y-axis represent?
What do the blue and orange columns represent?
Response: We apologize the lacking explanation. We added the sentences in the legend of Figure 5 as below:
“Figure 5. Expression profiles of six chrysanthemum genes, a) EXE, b) EXB1, c) LOB-1 and d) LOB-2, e) AS1-1, and f) AS1-2 observed in bud1 (blue) and bud2 (orange) during 2 days. Y-axis represents relative expression level against Actin gene. Asterisks indicate statistically significant differences based on the t test (P < 0.05) when comparing the results obtained on days 0 and 1 or days 1 and 2.”
We also added “Relative expression level (gene/Actin)” on Figure 5.
The context has no direct correlation. You can add before something sentences like that "Many genes in chrysanthemum may be related to growth and development."
Response: We thank the reviewer for the proposal. We would like to add the sentence as below:
“As we described in the Introduction, there were relatively many reports on transcrip-tome analysis in C. seticuspe [16] and even in chrysanthemum [9,10,17] and many genes in chrysanthemum may be related to growth and development.”
Please cite relevant literatures, especially about the genes including AS1, EBE, LOB, EXB1, you selected.
Response: we cited manuscripts to select Arabidopsis TFs relating axillary meristem formation and growth here.
- Yang, M.; Jiao, Y. Regulation of Axillary Meristem Initiation by Transcription Factors and Plant Hormones. Front. Plant Sci. 2016 7, 183.
- Wang, Y.; Jiao, Y. Axillary meristem initiation–a way to branch out. Curr. Opin. Plant Biol. 2018 41, 61-66.
We appreciate for your consideration of our paper.
Sincerely yours,
Katsutomo Sasaki, Ph. D.
Senior Researcher
-----------------------------------------------------------------------------------------NARO Institute of Floricultural Science (NIFS), National Agriculture and Food Research Organization (NARO), Fujimoto 2-1, Tsukuba, Ibaraki 305-8519, Japan
*E-mail: [email protected], Tel: +81-29-838-6822, Fax: +81-29-838-6841-----------------------------------------------------------------------------------------

Reviewer 3 Report (New Reviewer)
The manuscript presented by Tanaka and Sasaki is generally clear, well written and structured. Overall, the information presented represents valuable information regarding the axillary bud development in chrysanthemum. The release of the de novo transcriptome of chrysanthemum could be of interest for positive implications on both research and comparative study.
In my opinion, the manuscript is suitable for publication, although I have some comments and questions which I include here below:
Line 28: could be better ‘Chrysanthemum (Chrysanthemum morifolium Ramat.) is one of the….’
Line 63: ‘It is also known that different varieties have different responses to high and low temperatures and different tolerances of reaction temperatures.’ Please add reference;
Line 73: change ‘a phenomena’ with ‘a phenomenum’;
Line 82: ‘was also applied to’ …it's incomplete
Line 85: ‘than rice’ add ‘in’ before rice
Line 130: change ‘In case’ with ‘In the case’
Line 155: Su et al., 2019 in Current achievements and future prospects in the genetic breeding of chrysanthemum: a review. Horticulture Research, Volume 6, 2019, 109, https://doi.org/10.1038/s41438-019-0193-8 report as follows: ‘More recently, the genome information of two diploid Chrysanthemum species, C. nankingense and C. seticuspe, which are potential progenitors or model species of domesticated chrysanthemums, was published.’ I wonder why the C. nankingense information was not used in your transcriptome assembly. In any case, I would also recommend adding this work in your bibliography.
Line 292: change ‘In case’ with ‘In the case’
Line 307: change ‘regulates’ with ‘regulate’
Line 308: change ‘In case’ with ‘In the case’
Line 309: change ‘relating auxin and cytokinin’ with ‘relating to auxin and cytokinin’
Line 309: change ‘well studies’ with ‘well studied’
Line 327: ‘high-quality illumine reads’ change ‘illumine’ with with Illumina
Line 328: remove italic for ‘the’ in ‘the de novo’
Line 455: change ‘relating axillary bud’ with ‘relating to axillary bud’
My comments are in the review above.
Author Response
Dear. reviewer 3 Aug, 18th, 2023
We will answer in red below after your suggestion or identification.
The manuscript presented by Tanaka and Sasaki is generally clear, well written and structured. Overall, the information presented represents valuable information regarding the axillary bud development in chrysanthemum. The release of the de novo transcriptome of chrysanthemum could be of interest for positive implications on both research and comparative study.
In my opinion, the manuscript is suitable for publication, although I have some comments and questions which I include here below:
Line 28: could be better ‘Chrysanthemum (Chrysanthemum morifolium Ramat.) is one of the….’
Response: we revised.
Line 63: ‘It is also known that different varieties have different responses to high and low temperatures and different tolerances of reaction temperatures.’ Please add reference;
Response: We added the following references.
Nagasuga, K.; Yano, T.; Inamoto, K.; Yamazaki, H. Flowering and floret formation of summer-to-autumn flowering-standard type Chrysanthemums (Chrysanthemum morifolium Ramat.) under mist cooling during flower bud differentiation and developmental phase. Horticultural Res. (Japan) 2013, 12, 289-295.
Nakano, Y.; Takase, T.; Sugimoto, K.; Suzuki, S.; Tsuda-Kawamura, K.; Hisamatsu, T. Delay of Flowering at High Temperature in Chrysanthemum: Duration of Darkness and Transitions in Lighting Determine Daily Peak Heat Sensitivity. The Horticulture Journal 2020, 89, 602-608.
Line 73: change ‘a phenomena’ with ‘a phenomenum’;
Response: we revised.
Line 82: ‘was also applied to’ …it's incomplete
Response: we added the following words
“In 2020, long-read sequencing was also applied to the transcriptome analysis [11].”
Line 85: ‘than rice’ add ‘in’ before rice
Response: we revised.
Line 130: change ‘In case’ with ‘In the case’
Response: we revised.
Line 155: Su et al., 2019 in Current achievements and future prospects in the genetic breeding of chrysanthemum: a review. Horticulture Research, Volume 6, 2019, 109, https://doi.org/10.1038/s41438-019-0193-8 report as follows: ‘More recently, the genome information of two diploid Chrysanthemum species, C. nankingense and C. seticuspe, which are potential progenitors or model species of domesticated chrysanthemums, was published.’ I wonder why the C. nankingense information was not used in your transcriptome assembly. In any case, I would also recommend adding this work in your bibliography.
Response: We appreciate your suggestion to use public data. However, we also notice that these kinds of transcriptome data were more purpose driven data and not suitable for the other purpose, because assembled transcripts were in partial. Even in our dara, we could only 70% of C. seticuspe transcriptome. This means that our RNA-Seq data could not cover full transcriptome and condition specific gene expression affected to the contents. We described this contents in the “Discussion” as below.
“These transcripts, however, were created under specific conditions, such as florets, capitula, and flavonoid biosynthesis. Even in the study for axillary bud formation, the difference of the experimental conditions altered the relating TFs. In addition, no reports have published assembled transcript datasets. Since the purpose of this study was to determine the key genes for axillary bud development after cold treatment, expressed genes in the tissue under these conditions should be targeted.” On lines 370-377.
We would also like to cite this reference in the “Introduction”.
Line 292: change ‘In case’ with ‘In the case’
Response: we revised.
Line 307: change ‘regulates’ with ‘regulate’
Response: we revised.
Line 308: change ‘In case’ with ‘In the case’
Response: we revised.
Line 309: change ‘relating auxin and cytokinin’ with ‘relating to auxin and cytokinin’
Response: we revised.
Line 309: change ‘well studies’ with ‘well studied’
Response: we revised.
Line 327: ‘high-quality illumine reads’ change ‘illumine’ with with Illumina
Response: we revised.
Line 328: remove italic for ‘the’ in ‘the de novo’
Response: we revised.
Line 455: change ‘relating axillary bud’ with ‘relating to axillary bud’
Response: we revised.
We appreciate for your consideration of our paper.
Sincerely yours,
Katsutomo Sasaki, Ph. D.
Senior Researcher
-----------------------------------------------------------------------------------------
NARO Institute of Floricultural Science (NIFS), National Agriculture and Food Research Organization (NARO), Fujimoto 2-1, Tsukuba, Ibaraki 305-8519, Japan
*E-mail: [email protected], Tel: +81-29-838-6822, Fax: +81-29-838-6841
-----------------------------------------------------------------------------------------

Round 2
Reviewer 2 Report (New Reviewer)
Thank you for your efforts in addressing my comments and revising your manuscript. I look forward to seeing the key experimental evidence in your next research. There are a couple additional minor comments identified in the review of your revised manuscript. I hope you are willing to make these minor changes. I recommend the present version of the manuscript to plants for acceptance after minor changes.
1. R18 and R19, Chrysanthemum nankingense and Dendranthema grandiflorum should be italicized. Please check everything else carefully.
2. “real-time quantitative reverse transcription polymerase chain reaction (qRT-PCR)”, should it be abbreviated RT-qPCR or change the full name?
3. The statistical P should be italicized in Line 308 (P < 0.05).
4. Writing mistake in Line 295, “AS1-2 LOB” should add the word “and”, please modify it.
No comments.
Author Response
Dear. reviewer 2 Aug, 22nd, 2023
Thank you very much for your very favorable decision.
Your decision is an encouragement for our research.
I would like to respond to your points as follows.
- R18 and R19, Chrysanthemum nankingenseand Dendranthema grandiflorumshould be italicized. Please check everything else carefully.
> Our response
Thank you for pointing these out. We have corrected these according to your suggestion in red letters.
- “real-time quantitative reverse transcription polymerase chain reaction (qRT-PCR)”, should it be abbreviated RT-qPCR or change the full name?
> Our response
We had incorrectly written the official name of qRT-PCR. We therefore corrected “real-time quantitative reverse transcription polymerase chain reaction” to be correctly written as “quantitative real-time reverse transcription polymerase chain reaction (qRT-PCR)” Line 298 in red letters.
- The statistical P should be italicized in Line 308 (P < 0.05).
> Our response
We have corrected it according to your suggestion in red letters in Line 308 as (P < 0.05).
- Writing mistake in Line 295, “AS1-2 LOB” should add the word “and”, please modify it.
> Our response
We add the word “and” according to your suggestion in red letters in Line 295 and revised as “AS1-2, and LOB”.
We appreciate for your consideration of our paper.
With best regards,
Katsutomo Sasaki, Ph. D.
Senior Researcher
-----------------------------------------------------------------------------------------
NARO Institute of Floricultural Science (NIFS),
National Agriculture and Food Research Organization (NARO),
Fujimoto 2-1, Tsukuba, Ibaraki 305-8519, Japan
*E-mail: [email protected], Tel: +81-29-838-6822, Fax: +81-29-838-6841
-----------------------------------------------------------------------------------------

This manuscript is a resubmission of an earlier submission. The following is a list of the peer review reports and author responses from that submission.
Round 1
Reviewer 1 Report
In this study, transcriptome analysis of Chrysanthemum morifolium cultivar 'Jinba' was carried out to identify important genes for axillary bud development seen when moved from low-temperature treatment to normal cultivation temperature. Transcriptome analysis identified transcription factors that respond to low temperature and branching. However, the whole work is relatively simple, and there is no accurate description of the sample or other related details. The discussion lacks in-depth discussion of the six transcription factors.
Q1:The 3 months from cutting to vegetative growth of C. morifolium ‘Jinba’ was too long, and the growth status of the material was not clearly explained (line 97-99). The description like ' A sufficiently grown chrysanthemum ' is not accurate enough. The description of the material here is not accurate. It is recommended to use the number of leaves to indicate the material state.
Q2:The result and discussion need to be separated. Discussion should focus on key topics. Some content can be integrated into the result part.
Q3:What is a transcription factor? What is a transcript?Please confirm further!(line 353)
Q4:The discussion section lacks reading and exploration of related articles, and the identified transcription factors should be further explored.
Reviewer 2 Report
The manuscript is written well, however, this is purely in silico research, without any validation. The conclusion is not giving any proper insight and the title is also misleading.
I would like to see the proper validation of the conclusion drawn from the in silico works before considering this work for publication.
Reviewer 3 Report
The manuscript entitled "Detection of transcription factors relating to the axillary bud development after cold treatment in hexaploid Chrysanthemum morifolium using Arabiopsis information " submitted by Tanaka and Sasaki, provides some interesting information and the findings could be valuable for future research, there are a number of major weaknesses and drawbacks suggesting that the article fails to meet the standards for publication. Especially, the findings are mostly descriptive and not analyzed thoroughly. There are several points that have to be addressed by the authors towards improvement of the manuscript.
To be precise, major comments:
1. Text is not scientifically written. Language is not concise preventing from concentrating on the scientific content. Substantial editing is required by a native speaker.
2. Authors should verify the expression profiles of transcription factors using another method such as RT-qPCR. Also, this counts for other DEGs to verify the results of the RNA-seq experiments
3. An in-depth Gene Ontology analysis is very important for these studies and mostly needed to describe and discuss the results of Figures 2-4.
4. Authors should include the deposition number in SRA NCBI.
5. I cannot understand the purpose of the model in Fig. 6. Only four out of ~20 genes have been identified in this work.